# Effects of 5-Aminolevulinic Acid as a Supplement on Animal Performance, Iron Status, and Immune Response in Farm Animals: A Review

**DOI:** 10.3390/ani10081352

**Published:** 2020-08-04

**Authors:** Amin Omar Hendawy, Mostafa Sayed Khattab, Satoshi Sugimura, Kan Sato

**Affiliations:** 1Department of Biological Production, Tokyo University of Agriculture and Technology, Tokyo 183-8509, Japan; satoshis@cc.tuat.ac.jp; 2Department of Animal and Poultry Production, Damanhour University, Damanhour 22516, Egypt; 3Department of Dairy Science, National Research Centre, Dokki, Giza 12622, Egypt; ms.khattab@nrc.sci.eg; 4Laboratory of Animal Nutrition, Division of Life Sciences, Graduate School of Agricultural Science, Tohoku University, Sendai 980-8572, Japan

**Keywords:** aminolevulinic acid, broiler chicken/laying hen, dairy cow, feed additive, heme oxygenase-1, immunity/immunomodulator, iron status, nutritional supplement, performance, pig/sow

## Abstract

**Simple Summary:**

5-aminolevulinic acid is an amino acid that promotes the formation of heme—an essential constituent of hemoglobin. It has been recently used as a novel feed supplement to enhance the productivity of farm animals, but the current understanding of its effects on livestock is not clear. We systematically evaluated the literature for the effects of 5-aminolevulinic acid supplementation on animal performance, iron status, and immune response in farm animals. Extensive search of PubMed and Web of Science resulted into 16 eligible controlled trials. Findings revealed that iron status and immunity were most responsive to 5-aminolevulinic acid. Other parameters displayed hardly any tangible effect. Studies were highly heterogeneous (regarding species, dose, treatment duration, use of other supplements), which may limit the conclusion. Standard procedures and outcome measures are needed to confirm the benefits of 5-aminolevulinic acid. Attention should also be paid to any adverse effects.

**Abstract:**

Efforts directed toward enhancing animals’ productivity are focused on evaluating the effects of non-traditional feed additives that are safer than antibiotics, which have been banned because of their health hazards. Many studies used an amino acid that contributes to heme biosynthesis, known as 5-aminolevulinic acid (5-ALA), to promote the productivity of farm animals. However, these studies demonstrate inconsistent results. In order to develop a clear understanding of the effects of 5-ALA in farm animals, we comprehensively searched PubMed and Web of Science for studies evaluating 5-ALA effects on the performance, iron status, and immune response of different farm animals. The search retrieved 1369 publications, out of which 16 trials were relevant. The 5-ALA-relevant data and methodological attributes of these trials were extracted/evaluated by two independent researchers, based on a set of defined criteria. Samples were comprised of pigs, chickens, and dairy cows. The 5-ALA doses ranged from 2 mg to 1 g/kg of feed, and treatment duration ranged from 10 to 142 days. Overall, 5-ALA improved iron status in most studies and increased white blood cells count in 3 out of 10 studies, in addition to improving animals’ cell-mediated immune response following immune stimulation with lipopolysaccharide. Inconsistent findings were reported for growth performance and egg production; however, a combination of 10 mg/kg of 5-ALA with 500 mg/kg of vitamin C promoted the highest egg production. In addition, 5-ALA improved milk protein concentration. In conclusion, 5-ALA can enhance farm animals’ iron status and immune response; however, the heterogeneity of the reviewed studies limits the generalizability of the findings. Standard procedures and outcome measures are needed to confirm the benefits of 5-ALA. Attention should also be paid to any adverse effects.

## 1. Introduction

The ban set on the use of antibiotics that were mainly used to decrease energy loss and improve animals’ productivity forced those concerned with animal breeding to search for safer and better alternatives [1]. Researchers are obliged to explore novel animal production systems that provide high-quality products at low cost, such as the use of molecular nutritional techniques [2]. Some of the prevailing approaches to enhance animals’ productivity are based on diet enrichment with amino acids aimed to improve milk protein [3] or the supplementation of essential minerals, such as iron [4].

Unfortunately, direct addition of iron to feed may interfere with the digestion and absorption of other nutrients, such as the formation of insoluble complexes of P and Fe in the digestive tract (e.g., FePO_4_), which in turn leads to deficiency of both elements. As a result, several undesirable consequences may ensue, such as oxidative stress, bacterial infection, diarrhea, and reduced weight gain. Thus, fortifying feed with iron may not be an effective enrichment method, especially because iron toxicity (detected from a darker rumen color) has been reported with high-Fe doses [4]. In addition, it has been indicated that animals prefer drinking water without added Fe compared with water containing Fe sources with different anionic moieties [5].

Evolving research on feed additives involves new non-traditional ones, such as 5-aminolevulinic acid (5-ALA), a product of condensing succinyl-CoA and glycine through the catalytic activity of 5-ALA synthase [6]. After a chain of chemical interactions, 5-ALA changes into protoporphyrin IX—a heme precursor. Then, with the help of ferrochelatase, the porphyrin ring of protoporphyrin IX acquires an iron atom to finally produce heme, which is an essential constituent of hemoglobin [7].

Based on this understanding, it has been theorized that introducing 5-ALA as a dietary supplement to livestock can affect the synthesis of heme and positively influence the iron or hemoglobin status of animals [8]. 5-ALA has a wide range of applications in plants. On the one hand, it is used as an herbicide, insecticide, and growth-promoting factor [9]. On the other hand, it is also used to foster plants’ tolerance to salt and cold temperatures [10]. Furthermore, the literature documents an array of 5-ALA uses in the medical field, especially for tumor diagnosis and cancer treatment [11,12]. Similarly, 5-ALA has been introduced to farm animals as a dietary supplement to improve blood iron status [13,14,15], enhance immunity, increase resistance to disease [8,16,17], improve growth [18], and enhance egg production and quality [19,20,21], as well as milk composition [14,22,23]. However, available studies report mixed findings. To our knowledge, the benefits of 5-ALA supplement in farm animals have not been systematically reviewed. Though an existing narrative review theoretically enumerates the mechanism and benefits of 5-ALA [24], no sharp cutting decisions about the effectiveness of 5-ALA could be drawn from that review. To bridge the gap, the current study aims to systematically address the beneficial effects of 5-ALA for iron status, immune response, and animal performance.

## 2. Materials and Methods

This review was conducted according to the preferred reporting items for systematic reviews and meta-analyses (PRISMA) statement guidelines [25].

### 2.1. Literature Search

A five-stage, comprehensive search was conducted to identify eligible studies. In the first stage, two databases, namely PubMed and Web of Science, were searched to obtain all relevant 5-ALA studies that were published before March 2020. The used search strategy involved a combination of keywords: (5-aminolevulinic acid or δ-aminolevulinic acid or delta-aminolevulinic acid or aminolevulinic acid or aminolevulinate) and (cow or cattle or calves or heifer or buffalo or bull or steer or sheep or ewe or lamb or ram or goat or kid or duck or goose or geese or poultry or hen or broiler or chick or chicken or quail or pigeon or turkey or ostrich or rabbit or sow or swine or pig). The search was not restricted by language, date, or study type.

In the second stage, the total hits from both databases were pooled and duplicates were removed. The third stage involved screening of the retrieved articles by reading the article titles and abstracts. In the fourth stage, the full-length individual manuscripts were screened, and papers not satisfying the inclusion criteria (the next section) were excluded. In the fifth stage, to obtain additional data, we conducted a backward manual search of the reference lists of the selected articles.

### 2.2. Inclusion and Exclusion Criteria

To be eligible for inclusion, studies should have met the following inclusion criteria: (1) used 5-ALA as a feed supplement, either alone or in combination with other additives; (2) incorporated samples of any farm animal species; and (3) measured the outcome criteria of iron status, immune response, and animal performance. We excluded articles that were not published in English, conference proceedings, editorials, commentaries, and book chapters/book reviews.

### 2.3. Data Extraction

A special tabular form was designed for data collection. Two independent researchers screened all the selected studies to extract data related to the species, treatment and control groups, doses of the supplemented 5-ALA, combinations with other additives, duration of the trial, outcome measures, and the main findings.

### 2.4. Quality Criteria

To evaluate the methodological quality of the reviewed articles, we developed our quality criteria by referring to the Cochrane tool for reporting randomized trials, and also by adapting some relevant criteria from available systematic reviews in the field of animal production [26,27]. According to this method, studies are assessed for risk of different biases on a three-point rating scale: 2 = low risk (criteria properly performed/described), 1 = unclear risk (criteria not described), and 0 = high risk (criteria not met). The following criteria were considered for quality assessment:

(i) Selection bias: trials that used a randomization procedure were ranked as low-risk; when the method of randomization was not clearly described, they were ranked as unclear risk; however, when the study was non-randomized, it was ranked as high-risk.

(ii) Allocation concealment: trials that described procedures to conceal allocation were ranked as low-risk, while when allocation was not concealed trials were ranked as high-risk; when the procedure used for concealment was not reported, it was ranked as unclear risk.

(iii) Performance bias: when treatment was conducted by experimenters who were blinded to the study aim/hypothesis, the study was ranked as low-risk; when experimenters were not blinded to the study aim/hypothesis, the study was rated as high-risk; however, when blinding of experimenters was not reported, it was ranked as unclear risk.

(iv) Detection bias: when laboratory analyses were evaluated by examiners who were blinded to the study aim/hypothesis, studies were rated as low-risk; when examiners were not blinded to the study aim/hypothesis, the studies were rated as high-risk; however, when blinding was not clearly reported, these studies were rated unclear risk.

(v) Reporting bias: when all outcomes described in the methods section were reported in the results (and reports were consistent between the abstract, results, and discussion) the studies were rated as low-risk; studies with missing outcomes were rated unclear risk; whereas studies with inconsistent reports between the abstract, results, and discussion were rated as high-risk.

(vi) Sample size: the number of animals per group was determined according to the type of species—pigs and cows (up to 30 was rated as high-risk; more than 30 was rated as low-risk), and birds (up to 50 was rated as high-risk; more than 50 was rated as low-risk).

(vii) Breed or genetic line: when breed or genetic line was detailed, trials were rated as low-risk, but when not described, trials were rated as high-risk.

## 3. Results

### 3.1. Literature Search

Of all the obtained publications (1369), 1347 studies were excluded after the initial screening of titles and abstracts: 250 papers were duplicated, 52 papers were not in English, 45 papers were review papers and meta-analyses, and 1000 articles were irrelevant. We examined the full text of 22 articles, and 6 papers were further excluded, as they were repeated reports from the same samples. In total, 16 studies were included in this review (Figure 1). The selected articles were published between 2006 and 2019.

### 3.2. Description of Included Studies

Sixteen controlled trials were included in this review. The duration of 5-ALA supplementation ranged from 10 days to 69 days, with the exception of one long-term study in which 5-ALA was supplemented to sows during gestation and lactation periods (including the weaning-to-estrus intervals) [28]. The dose of 5-ALA supplemented in these studies varied from 2 mg to 1 g 5-ALA/kg of feed. Seven studies included experimental groups that were fed 5-ALA combined with vitamin C [14,15,20], antibiotics [17], chito-oligosccharide [29], oriental medicinal plants [30], and iron injection [28].

Out of 16 studies, nine used pigs’ samples (three studies had sow subjects, while six studies were comprised of weanling or finishing pigs). The remaining seven studies included chicken samples—three studies involved broiler chickens and three studies included laying hens. One study included dairy cows.

All studies measured at least one indicator of growth performance, such as average daily gain (ADG), average daily feed intake, feed efficiency (G/F), feed conversion ratio (F/G), body weight (BW), BW at birth and weaning, born-alive litter size, and nutrient digestibility, such as dry matter, nitrogen, and energy digestibility. Moreover, seven studies assessed production performance. Four studies examined milk composition, e.g., milk fat, milk protein, milk casein, milk glucose, milk lactose, total solid, and Fe concentration [14,22,23,28]. Egg production and quality were also assessed in three studies: egg weight, shell breaking strength, shell thickness, Haugh unit, yolk color unit, yolk index, shells color, albumin height, and Fe concentration in yolk [19,20,21].

The effect of 5-ALA supplementation on iron status was studied in 14 studies that used five indices: hemoglobin, hematocrit, Fe, total iron binding capacity (TIBC), and red blood cells (RBC). In two studies, iron was measured in plasma, and in the rest of the studies it was measured in serum. Herein, iron status refers to serum iron levels, unless otherwise specified.

In this review, we considered indices reflecting different levels of defense mechanisms of the immune system: white blood cell (WBC) count, differential count, and cell-mediated immunity, which involves soluble proteins and bioactive small molecules released by the activated cells, such as cytokines, as well as the membrane-bound receptors and cytoplasmic proteins that bind to molecular patterns expressed on the surfaces of invading pathogens. Hence, the immune response in the evaluated studies was addressed through an estimation of WBC count and lymphocyte count, as well as measurement of tumor necrosis factor (TNF)-α; insulin-like growth factor (IGF)-1; haptoglobin; plasma cortisol [8]; cluster of differentiation antigen positive cells 2, 4, and 8 (CD2+, CD4+, CD8+, respectively); the ratio of CD4+ to CD8+ cells; B-cells; major histocompatibility complex classes I and II [17]; rates of phagocytosis in blood mononuclear cells (MNC); mitogen concanavalin A and phytohemagglutinin-induced proliferation of blood MNC [16,22]; levels of cluster of differentiation 3 (CD3) mRNA in the spleen; plasma thiobarbituric acid reactive substances (TBARS); plasma ceruloplasmin concentration; expression of interferon-γ; inducible nitric oxide synthase (iNOS); interleukin (IL)-6 and TNF-like ligand 1A mRNA; levels of IL-2; toll-like receptor 2, 4, and 7 mRNA in the spleen during *Escherichia coli* lipopolysaccharide (LPS) stimulation; ceruloplasmin oxidase [16]; differential count of WBC [13]; immunoglobin G (IgG) [8,14,29]; and the weight of immune organs, such as spleen, bursa of Fabricius, liver, and the thymus [15,16,31]. Readers are encouraged to refer to Appendix A for further details.

The quality of the selected articles was evaluated according to the aforementioned criteria. The maximum achieved quality score was 11 points [15,17,18,19,21,31], while the minimum score was eight points [14,30] (Appendix A).

Figure 2 and Figure 3 show that except for one study [13], all included studies used randomization procedures; thus, they were low-risk for selection bias. Not a single article indicated or described the use of any procedure of allocation concealment, blinding of treatment performers, (in farms) or blinding of analyses performers (in laboratories). All papers described the breed/animal genetic line. Reporting bias was noticed in 12.5% of the papers [14,30]. Less than half the studies (44%) used more than 30 pigs/cows or 50 birds [13,15,17,18,19,21,31]. It is worth noting that two studies, comprising four experiments, were rated as high-risk for sampling bias, because animal numbers in some groups were small, though some groups had adequate numbers [16,28].

### 3.3. Effects of 5-ALA on Animal Performance

Table 1 summarizes effects of 5-ALA in different farm animals. Overall, no significant effects of 5-ALA supplementation on growth performance were observed, except in two studies that reported improved ADG and G/F [14,18]. Significant increases were reported for BW [14,16,18,28], born-alive litter size, and BW at birth and weaning [28]. Regarding nutrient digestibility, only two studies reported significant positive effects of supplementation of 5-ALA in different dietary levels for dry matter [8,18] and nitrogen digestibility at the end of treatment [8].

Concerning production performance, 5-ALA supplementation improved milk protein [14,22,23], milk fat [14], milk casein [22], and Fe concentration in milk [28], while milk glucose, milk lactose, and total solids were unaffected. Similarly, 5-ALA supplementation improved egg production [20,21] and quality, such as egg weight [21], Haugh unit [19,20,21], yolk color unit [20,21], eggshell color, Fe concentration in yolk [20], and egg yolk index [19]. However, there were no changes in some of the aforementioned studies in egg production, egg weight, eggshell breaking strength, eggshell thickness, yolk color unit, and albumin height.

### 3.4. Effect of 5-ALA on Iron Status

Out of the 13 studies that measured hemoglobin concentration, 5-ALA improved hemoglobin in 8 out of 15 results (e.g., in sows and their piglets). As for hematocrit concentration, three out of seven results revealed some 5-ALA related improvements. Regarding iron concentration, 5-ALA improved plasma and serum Fe concentration in 11 out of 15 results. In relation to TIBC concentration, only 3 of the 12 results witnessed an increase of TIBC concentration because of 5-ALA supplementation. However, RBC count was improved in 8 out of 12 results as a result of the use of 5-ALA.

### 3.5. Effect of 5-ALA on Immune Response

Supplementation of 5-ALA was associated with improved IGF-1 (2 h) and WBCs (12 h) post-challenge by LPS [8]. Also, 5-ALA increased levels of CD2+, CD8+, B-cells, major histocompatibility complex classes I and II [17], the rate of phagocytosis and mitogen-induced proliferation of peripheral MNC [16,22], CD3 mRNA in the spleen, and plasma TBARS [16]. On the other hand, dietary use of 5-ALA reduced plasma cortisol and TNF-α concentration at 2 h post-challenge by LPS [8]. Similarly, another trial revealed noticeable decreases of IL-2 levels, plasma ceruloplasmin concentration (24 h), expression of interferon-γ, iNOS, IL-6, and TNF-like ligand 1A mRNA (3 h) after LPS injection [16]. However, findings were mixed, as some studies showed no changes in lymphocyte and haptoglobin post-challenge by LPS [8]; CD4+ and the ratio of CD4+ to CD8+ [17]; interferon-γ mRNA expression; and levels of CD3, IL-2, and toll-like receptors 2, 4, and 7 mRNA in the spleen during LPS stimulation [16]. There was no change in the differential count of WBCs in one study [13]. While the number of WBCs and the percentage of lymphocytes were examined in 10 studies, only three studies reported an increase of WBCs [8,21,22] and percentage of lymphocytes [20,21] as a return for 5-ALA supplementation. Meanwhile, IgG was measured in three studies, and it was increased in two of them [14,29]. Captured in a sole study, the plasma ceruloplasmin oxidase activity of piglets decreased due to 5-ALA supplementation [28]. Regarding the relative weight of the immune organs, mixed findings were indicated. In one study, 5-ALA increased the weight of both the spleen and the bursa of Fabricius [31]. On the contrary, two other studies indicated no change in the weight of the spleen, the liver, the bursa of Fabricius, or the thymus gland [15,16] (see Appendix A for further details).

### 3.6. Effects of 5-ALA Compared with Other Alternative Treatments

In three studies using vitamin C as an alternative treatment, 5-ALA alone had superior effects to vitamin C alone and vitamin C combined with 5-ALA on milk composition (in sows), serum Fe levels, hemoglobin, and RBC count, both in poultry and porcine species (Appendix A). Meanwhile, vitamin C on its own had no effect on any indicators of Fe status, except for hematocrit (in one study) [15]. Interestingly, piglets of sows receiving a combination of vitamin C with 5-ALA exhibited significant increases in BW at weaning and ADG compared with other groups [14]. Nonetheless, a combination of 5-ALA and vitamin C had a better effect on lymphocytes than each treatment on its own in poultry [20]. In laying hens, solo 5-ALA treatment significantly improved albumin height, yolk color unit, eggshell color, and Haugh unit, while vitamin C alone improved only the last three parameters. Strangely, combinations of 5-ALA and vitamin C demonstrated no effect on any of these egg qualities, though eggshell thickness and iron concentration in yolk significantly increased [20].

Two studies compared the effects of antibiotics (apramycin) [17] and chito-oligosccharides [29] with that of 5-ALA on iron status in weanling pigs. Neither antibiotics nor chito-oligosccharides could express any positive effect except for lymphocyte count (only chito-oligosccharides). On the other hand, solo 5-ALA increased Fe levels, TIBC, RBC [29], hemoglobin, and hematocrit [17]. In the meantime, treatments involving 5-ALA in combination with antibiotics or chito-oligosccharides could not produce any significant effect. 5-ALA with or without antibiotics was associated with an increase of the levels of CD2+, CD8+, B-cells, and major histocompatibility complex classes I and II [17]. In a single study, inorganic iron (FeSO_4_) was used as a positive control, and authors compared the effects of 5-ALA and iron injection (iron dextran) on iron status and growth performance in young pigs. Both 5-ALA and iron injection significantly increased Fe, hematocrit, hemoglobin, and BW at weaning. Animals receiving a combination of 5-ALA and iron injection have more positive effects than animals receiving only a single treatment [28]. Further details are shown in Appendix A.

### 3.7. Mechanism of Action of 5-ALA

Heme formation process initiated by 5-ALA starts in the cytoplasm, where 5-ALA sequentially generates porphobilinogen, hydroxymethylbilane, uroporphyrinogen III, and eventually coproporphyrinogen III. Coproporphyrinogen III translocates into the mitochondria, where it gets metabolized to protoporphyrinogen IX and protoporphyrin IX, into which iron is inserted through a ferrochelatase-catalyzed reaction, resulting in heme synthesis [6]. It seems that the action of 5-ALA is adaptive, i.e., it varies in response to changes in the internal microenvironment of the body. In this respect, 5-ALA enhanced the levels of hemoglobin, RBCs, and serum Fe concentration in young pigs [8,14,18], which may suggest that the effect of 5-ALA on heme generation occurs under conditions of iron deficiency. This is because young pigs commonly exhibit iron deficiency for a number of reasons: low hepatic iron stores in newborn pigs, low iron concentrations in the milk of the sows [32], and rapid increase in RBC volume and body mass [33]. In support of this view, research shows that heme has a feedback effect on 5-ALA synthase, which gets downregulated in conditions involving increased heme concentrations, resulting in suppression of heme synthesis [31,34].

From another perspective, levels of endogenous protoporphyrin induced by systemic administration of 5-ALA vary in different tissues of the body—protoporphyrin was not detected in muscle, but relatively higher levels (18%) were detected in the liver, a main detoxifying organ, and much higher levels were detected in tumor tissues. As a result, protoporphyrin suppressed energy phosphate metabolism, causing tumor adenosine triphosphate levels to drop to near zero, leading to high necrosis and the inhibition of tumor growth. In that study, the activities of certain selected enzymes involved in heme biosynthesis had no association with porphyrin concentrations in different tissues [35]. In the current review, 5-ALA had no effect on heme synthesis in some studies (e.g., involving dairy cows); however, it boosted immune functioning [22]. Evidence associates inflammation and oxidative stress that foster senescence with levels of iron overload-related mitochondrial ferritin. As an adaptive response, the master antioxidant molecule, nuclear factor erythroid 2 (NRF2, gets activated in order to upregulate levels of phase II antioxidants, such as heme oxygenase-1 (HO-1). Phase II antioxidants get activated in pathological conditions as a defense against insults induced by cellular toxins. Therefore, phase II antioxidants trigger an array of cellular activities that aim at restoring cellular structure and function [36]. In this respect, the above-reviewed studies signify an immunomodulatory effect of 5-ALA. It seems that the biological activities of 5-ALA, such as being an antioxidant, anti-inflammatory, and immunomodulator underlie the reported effects (Figure 4). Herein, we explore the literature for the molecular actions that promote the positive effects of 5-ALA.

A wealth of studies reports a potent antioxidant effect of 5-ALA [6,37]. In details, 5-ALA promotes the production of internal antioxidants, such as HO-1 [6,37]. HO-1 promotes the production of other antioxidant molecules, such as biliverdin, ferrous iron, and carbon monoxide. The action underlying this activity is promoting the degradation of heme [37,38].

Similar to the findings reported by Sugiyama et al. [38], we detected a slight increase of TNF-α in peripheral blood MNC (obtained from untreated dairy cows) treated with 5-ALA in the absence of LPS, while co-treatment with LPS resulted in a significant reduction in the mRNA expression of TNF-α (unpublished data). The literature documents that mild levels of free radicals, such as reactive oxygen species (ROS) and reactive nitrogen species (RNS), promote optimal biological functioning. For example, ROS/RNS support the production of antioxidants through the activation of NRF2, and they accelerate the degradation of tumor cells and enhance autophagy and intracellular defenses against pathogens [39,40]. It is well-known that LPS treatment elevates ROS and RNS production [39]. Meanwhile, the production of HO-1 increases following treatment with a range of stress stimuli [6]. In addition to their antioxidant activities, HO-1 and NRF2 also suppress inflammation through inhibition of the key inflammatory pathway: nuclear factor kappa light chain enhancer of activated B cells (NF-kB) [40,41,42].

Therefore, the anti-inflammatory effects of 5-ALA may be a mere translation of its antioxidant effects. In other words, pretreatment with 5-ALA would induce a mild degree of oxidative stress, resulting in cellular pre-conditioning to ROS and RNS. Subsequent treatment with LPS would result in the activation of cellular signaling necessary for cell protection against the high emission of free radicals and inflammatory mediators in pre-conditioned cells, compared with untreated cells [40]. The cytoprotective role demonstrated by HO-1 is documented in several studies. In this regard, HO-1-deficient cells display limited stress resistance [43], and people expressing deficiency in genes involved in the production of HO-1 demonstrate higher production of inflammatory markers [44]. Likewise, an in vivo investigation involving treatment of HO-1-deficient rodents with endotoxins is associated with the body flooding with cytokines, and necrosis in key organs involved in immune protection, such as the liver and the spleen [45]. Finally, the 5-ALA mechanism of actions seem to be adaptive to internal needs of body environment, promoting heme synthesis under conditions of deficiency and promoting proper cellular housekeeping via enhancement of the activity of internal antioxidants. However, more studies are needed to examine the plausibility of these pathways.

## 4. Discussion

To our knowledge, this is the first attempt to systematically investigate the benefits of 5-ALA in farm animals. To determine its potential effects on animal performance, iron status, and immune response, we summarized data from 16 studies that used 5-ALA as a supplement in various farm animals. Except for milk protein, animal performance indices revealed few significant benefits of 5-ALA for growth performance, or for egg production and quality. Meanwhile, mixed findings were reported, with some significant effects observed for indices of iron status and immune response. However, studies indicating significance exhibited some heterogeneity with regard to 5-ALA doses; duration of supplementation; and animal species/breed, sex, and age. Generalizability of the findings is limited because of the heterogeneity of the techniques and outcome measures; in addition, samples consisted only of pigs, chickens, and dairy cows.

Despite the fact that an existing review of 5-ALA as a potential feed additive had been conducted by Cho and Kim, that review is narrative and theory-based—in a few instances, it addressed examples of the effects of 5-ALA from few, subjectively selected studies [24]. We located another narrative review of the role of 5-ALA in the regulation of iron metabolism in animals, but it is in Chinese [46]. Therefore, our attempt to systematically examine the benefits of 5-ALA in farm animals uniquely expands the existing knowledge about the effects of 5-ALA in farm animals.

In theory, 5-ALA does not directly influence the animal bodies; it has a role in heme synthesis, through which heme or hemoglobin content improves [24]. Some effects of 5-ALA on oxidative stress have been indicated. In one study, plasma ceruloplasmin oxidase activity—which increases as a result of the deficiency of antioxidant enzymes during iron deficiency—decreased in response to a dose of 90 mg/kg of 5-ALA, both with and without iron injection [28]. On the other hand, another study revealed an adverse effect of 5-ALA over long-term use, where plasma TBARS, a marker of lipid peroxidation and oxidative stress, increased in broiler chickens treated by 5-ALA compared with the control groups and chickens treated by the same 5-ALA dose for 10 days [16]. However, TBARS concentration in the loin meat of swine treated by a combination of 5-ALA and oriental medicinal plants was significantly lower than in the control group [30].

No overall improvement was noticed in most growth indices. However, the highest significant effect on growth performance was noticed in few studies that used high doses (500 mg and 1 g 5-ALA/kg of feed) of 5-ALA [18] and the long-term use of 10 mg 5-ALA/kg of feed [16]. Furthermore, some significant changes were noted at certain stages of development—e.g., ADG significantly improved during the first week of life in piglets of sows fed on 5-ALA [28], while two studies showed an increase in ADG on day 21 [8] and from 2 to 5 weeks of treatment [29].

Pertaining to the remaining outcome indicators, data from the reviewed studies did not follow a certain pattern. For example, milk protein and fat improved with supplementation of 10 mg/kg of 5-ALA for 28 days [14], whereas milk Fe improved with supplementation of 90 mg/kg of 5-ALA to sows during the gestation and lactation periods [28]. Interestingly, the use of trivial doses of 5-ALA (2 and 4 mg 5-ALA/kg) resulted in improvement of egg production [21]. Nonetheless, administration of 10 mg/kg of 5-ALA combined with 500 mg/kg of vitamin C gave the highest egg production and improved egg qualities [20]. Blood-related outcomes, especially Fe, seemed to be generally improved in both standalone and combined different doses of 5-ALA for different durations. As hypothesized, 5-ALA as feed additive turned out to be more effective than iron supplement for enhancing indices of Fe status. In fact, Fe as a feed additive has failed to maintain indices of Fe status in pigs [47]. Similarly, standalone administration of 5-ALA produced immunity improvement similar to the combinations, or even had better effects. Accordingly, it is fair to say that if the use 5-ALA is intended to improve animals’ iron status and immunity, then for cost-effectiveness it should be used solo.

It is noteworthy that in studies that measured WBC counts and lymphocyte counts (10 studies), there was a considerable increase in only three studies. It is well-known that cellular immunity is activated during times of infection [48]; hence, the reported lack of effect of 5-ALA on WBC count could be because involved animals were infection-free. Notably, indices of the immune response other than cellular immunity (soluble proteins and bioactive molecules) were estimated in a few studies, and similar to the cellular immune response, few overall improvements were reported. On the other hand, in two studies LPS was used to mimic the effect of pathogen invasion, and the pro-inflammation markers noticeably improved in response to LPS stimulation only in the 5-ALA groups compared with the control groups: in one study, the level of IGF-1 increased, whereas TNF-α and cortisol decreased 2 h after LPS injection [8], and plasma ceruloplasmin, interferon-γ, iNOS, IL-6, and TNF-like ligand 1A decreased 3 h after LPS injection [16]. This finding is supported by results from an existing review that reported that 5-ALA regulates the inflammatory response in several conditions in humans [6].

Most improvements in hemoglobin, Fe, and RBCs were noticed in young pigs directly supplemented with 5-ALA [8,13,17,18,29], or those nursing sows supplemented with 5-ALA as a result of increased milk content of iron [14,28]. In addition, 5-ALA increased milk protein both in sows and dairy cows [14,22,23], which indicates that 5-ALA-supplemented animals can be a source of naturally iron-fortified milk. Furthermore, 5-ALA increased egg yolk iron concentration, Haugh unit, eggshell color, egg yolk index, and egg yolk color unit [19,20,21]. Accordingly, 5-ALA may be most beneficial in young pigs, lactating animals, and laying hens. Additionally, 5-ALA expressed positive effects on the immune response, both under normal conditions [17,22] and under infection-like conditions involving challenge with LPS [8,16]. Therefore, it might be recommended to supplement animals at risk for infection (e.g., around parturition) with 5-ALA to attain prophylactic effects.

In short, results from studies involving the use of high doses as well as long-term use of low doses of 5-ALA seem to be mostly positive. Nevertheless, from the available data we cannot clearly recommend a dose, combination, or duration of feeding. Peculiarly, the use of combinations yielded mixed results; for instance, use of 10 mg/kg of 5-ALA with 500 mg/kg of vitamin C for 28 days in mother pigs immediately postpartum significantly improved piglet BW and ADG [14]. In contrast, the same combination had no effect on the growth performance of broilers [15]. We can conclude that some species might benefit better than others from 5-ALA combinations. This may be related to major differences in stomach structure and metabolism between different animal species.

Our findings should be considered within the boundaries of the heterogeneity, as well as the validity threats of the included studies. Although most studies used randomization, no studies mentioned that the person who conducted the randomization was blinded to the treatment, which implies unclear or high risk of selection bias, due to lack of allocation concealment. Also, not in a single study were experimenters who conducted animal treatment either in farms or laboratories blinded to treatment assignments, which can give rise to performance and detection biases. We also noted some instances of reporting errors, where values in tables were not consistent with the ones indicating the same outcomes in the text and abstract. In brief, within the context of the detected methodological flaws and the obvious variations in procedures, the spotted significant effects cannot be fully attributed to use of 5-ALA, because variations in animals’ characteristics and the environment can affect animals’ response to treatment. Moreover, the noted non-significance in most outcome indicators is somewhat alarming, because it puts the cost-effectiveness of 5-ALA in question. Future cost-effectiveness studies that use standardized procedures and unified outcome measures are needed to clarify the effects of 5-ALA in farm animals.

This review has some strengths, given that PubMed and Web of Science databases were extensively searched through a comprehensive list of key words. It also has some limitations. Three non-English studies were excluded, and the available studies exclusively were comprised from samples of pigs, chickens, and dairy cows, which limit the generalization of the findings. In addition, some outcomes were addressed in a limited number of studies—e.g., most indices of the cell-mediated immune response and cortisol hormone were measured only in single studies [8,16,17]. Noticeably, the majority of studies reported in this review were produced by a single group of researchers. In addition, the enormous heterogeneity between studies with regard to doses and outcomes measured, as well as the poorly reported findings, limited our potential for meta-analysis.

## 5. Conclusions

Use of 5-ALA as a dietary supplement had a significant beneficial effect on iron status and immune response. However, growth performance witnessed almost no change in response to 5-ALA. Nevertheless, 5-ALA would produce the most beneficial effects in young pigs by correcting anemia, in lactating animals by increase milk protein and iron contents, and in laying hens by improving egg production and egg quality. These effects may be attained in a short time (within 2 or 3 weeks of the commencement of treatment). According to cited studies, the minimal effective doses of 5-ALA are 2 mg/kg feed in laying hens; 10 mg/kg feed in broilers, weanling pigs, and dairy cows; and 500–1000 mg/kg feed in sows.

Although only a single study noted oxidative stress with long-term use, this is alarming, and future studies should address this outcome. Generally speaking, reviewed studies were limited in number and quality; outcomes were numerous, which made it difficult to group and report tangible outcomes. Standard procedures and established evaluation measures are essential for future studies, in order to fully reveal the true effects of 5-ALA supplementation.

## Figures and Tables

**Figure 1 animals-10-01352-f001:**
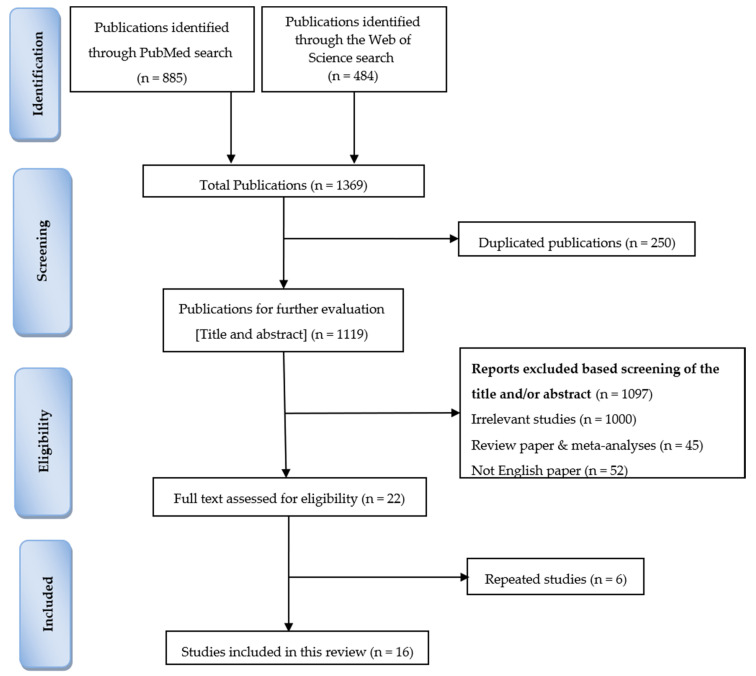
Summary of the search strategy based on the preferred reporting items for systematic reviews and meta-analyses (PRISMA) statement guidelines.

**Figure 2 animals-10-01352-f002:**
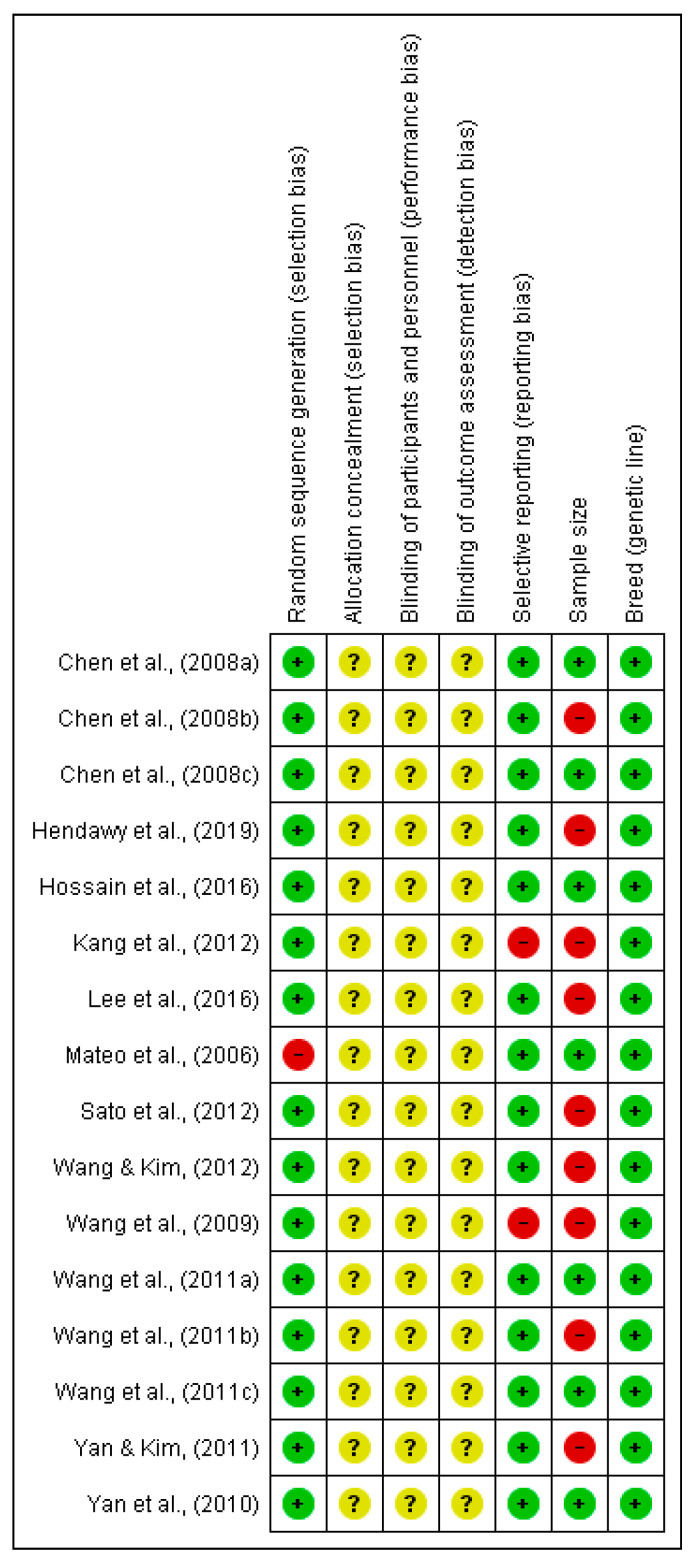
Quality assessment of each included studies.

**Figure 3 animals-10-01352-f003:**
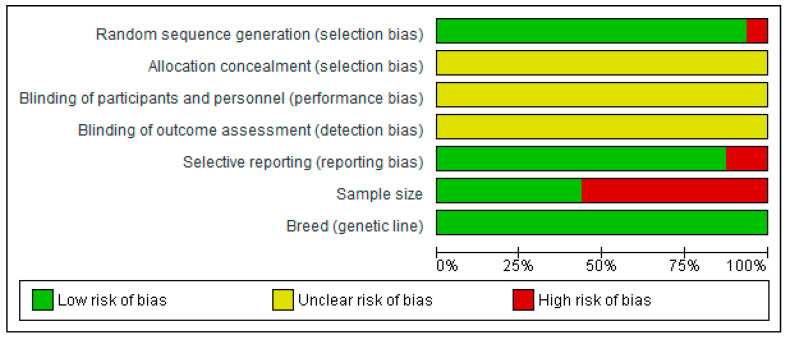
Overall quality assessment summary.

**Figure 4 animals-10-01352-f004:**
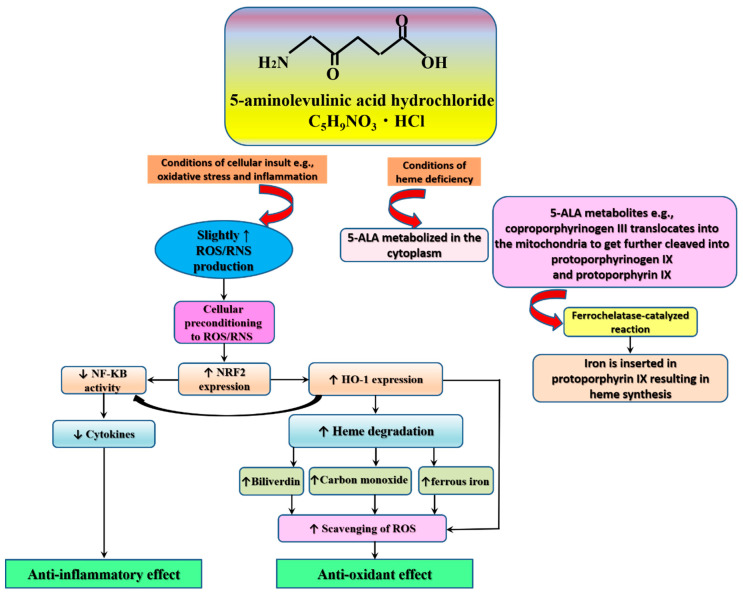
The possible mechanism of action of 5-aminolevulinic acid. Abbreviations: ROS, reactive oxygen species; RNS, reactive nitrogen species; NRF2, nuclear factor erythroid 2; HO-1, heme oxygenase-1; NF-kB, nuclear factor kappa light chain enhancer of activated B cells.

**Table 1 animals-10-01352-t001:** Effects of 5-aminolevulinic acid (5-ALA) supplementation on different farm animals.

Species	Growth Performance	Production Performance	Immune Function	Iron Status	Ref.
Broiler chickens	0	ND	+	0	[31]
Broiler chickens	+	ND	+	ND	[16]
Broiler chickens	0	ND	0	+	[15]
Laying hens	ND	+	0	0	[19]
Laying hens	ND	+	+	+	[20]
Laying hens	ND	+	+	+	[21]
Weanling pigs	0	ND	+	+	[8]
Weanling pigs	+	ND	ND	+	[18]
Weanling pigs	0	ND	0	+	[13]
Weanling pigs	0	ND	+	+	[17]
Weanling pigs	0	ND	+	+	[29]
Fattening pigs	0	ND	ND	ND	[30]
Pregnant and lactating sows▲	+	+	ND	+	[23]
Pregnant and lactating sows▲	+	0	ND	+	[28]
Parturient sows▲	+	+	+	+	[14]
Lactating dairy cows	ND	+	+	0	[22]

+: Positive effect; 0: no effect; ND: not determined; ▲: growth performance indices were measured in progenies.

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
