# Peer review of "Effects of 5-Aminolevulinic Acid as a Supplement on Animal Performance, Iron Status, and Immune Response in Farm Animals: A Review"

_animals, 2020, doi:10.3390/ani10081352_

Round 1

Reviewer 1 Report

General Comments:

As a literature review, this manuscript has rigorous logic in general. It aims to systematically illustrate the effects of 5-Aminolevulinic acid (5-ALA) for animal performance, iron status and immune response on farm animals such as piglets, laying hens, dairy cows and so on. And authors make a great deal in this manuscript about the way they search literature, their criteria of inclusion and exclusion and their quality criteria of the literature to ensure the authority and reliability of this review. As there are no sharp cutting decisions about the effectiveness of 5-ALA before, this manuscript has bridged the gap. It is valuable and well organized. However, the following points need clarification and should be taken into consideration. Besides, as I am not a professional or native English user, I can only suggest authors to find professional or native English users to check over this manuscript.

Major Comments:

Abstract:

There is a large part about analysis and summary of the literature in text. So, I suggest that it should add some appropriate supplements about the analysis in Abstract.

Material and Methods:

I think Table 1 can be more easily to read and understand after some modifications. For example, do not use abbreviation of the animal species and category in the “A” row of Table 1 so that the readers can get the useful information quickly. And it is better to rank the “A” row by the first letter, which could make this table more systematic.

And there has some problem about the abbreviation in text. If the phrase has a frequently-used abbreviation, it is suggested to explain and use. For example, the authors have use “average daily feed intake” as “ADFI” and “body weight” as “BW” in Table 1, so I think in line 153 “average daily feed intake” and “body weight” should also supply their abbreviations. Also in line 197 and line 339, replace “body weight” to “BW”, in line 230 “immunoglobin G” to “IgG”, in line 329 “insulin-like growth factor-I” to “IGF-1”, in line 330 “tumor necrosis factor-α” to “TNF-α” and so on. As I do not list all situations perhaps, I suggest authors to check over this manuscript carefully to make sure that all abbreviations are used properly.

In 3.3, line 195-200, I suggest authors to give a specific description about the effect of 5-ALA in animal performance. For example, it is better to descript whether it has a positive or negative effect, it increases or decreases some index and so on.

In 3.3, 3.4 and 3.5, readers may feel confused if the conclusions from different studies are conflicting. Authors could give a short explain or descript the condition of studies briefly, as you can explain your detailed analysis at the Discussion.

In line 251, I think that the ”nitrogen species” is not current. It should be “reactive nitrogen species (ROS)”. Besides, the “NS” in line 254 and line 261 should also be modified to “RNS”. And in Figure 4, it is not proper to ignore RNS (or NS?) cause it has been describe before.

In line 252, what does the “they” means? Does it mean “the free radicals such as ROS and RNS”? I think it should be clear.

In 3.6, the last paragraph (line 259-269), the mechanism of antioxidant effects of 5-ALA should be stressed. For example, you can explain the relation between 5-ALA and the antioxidant factors such as HO-1 in that paragraph briefly. That will make reader more easily to understand.

Discussion:

In line 306-307, it is better to descript the content of the “improvement” detailed.

In the line 362, the sentence “cortisol hormone were measured only in single studies”, I think it should be confirmed whether there need to cite references.

Minor concerns:

1.Line 148: replace “chitooligosccharide” to “chito-oligosccharide”.

2.Line 149: It should have a blank at the beginning of the paragraph.

3.Line 239: replace “5ALA” to “5-ALA”

4.Line 330: replace “two hr” to “2 hr”. Line 332: replace “three hr” to “3 hr”. I suggest that it is proper to use Arabic Number uniformly in this manuscript.

5.Line 331: replace “IL 6” to “IL-6”.

Author Response

Reviewer 1

Major Comments:

Abstract:

  1. There is a large part about analysis and summary of the literature in text. So, I suggest that it should add some appropriate supplements about the analysis in Abstract.

Response to comment 1: We agree with the reviewer; this part was missing in the abstract. We briefly reported on this section in this revised version (page 1, lines 34-36).

Material and Methods:

  1. I think Table 1 can be more easily to read and understand after some modifications. For example, do not use abbreviation of the animal species and category in the “A” row of Table 1 so that the readers can get the useful information quickly. And it is better to rank the “A” row by the first letter, which could make this table more systematic.

Response to comment 2: Thank you so much for this comment. We modified this table by including full terms for animal species, doses, treatment duration, measurement outcomes, and results. Further modifications were also recommended by reviewer 2 to make the table briefer and more readable. The detailed table is kept as a supplement in this version in case readers would look for extra details.

  1. And there has some problem about the abbreviation in text. If the phrase has a frequently-used abbreviation, it is suggested to explain and use. For example, the authors have use “average daily feed intake” as “ADFI” and “body weight” as “BW” in Table 1, so I think in line 153 “average daily feed intake” and “body weight” should also supply their abbreviations. Also in line 197 and line 339, replace “body weight” to “BW”, in line 230 “immunoglobin G” to “IgG”, in line 329 “insulin-like growth factor-I” to “IGF-1”, in line 330 “tumor necrosis factor-α” to “TNF-α” and so on. As I do not list all situations perhaps, I suggest authors to check over this manuscript carefully to make sure that all abbreviations are used properly.

Response to comment 3: The reviewer is absolutely right. We revised the entire manuscript and made sure that all abbreviations were used properly.

  1. In 3.3, line 195-200, I suggest authors to give a specific description about the effect of 5-ALA in animal performance. For example, it is better to descript whether it has a positive or negative effect, it increases or decreases some index and so on.

Response to comment 4: As the reviewer indicated, key effects of 5-ALA were specified e.g., positive effect/increase (page 7, lines 209-214).

  1. In 3.3, 3.4 and 3.5, readers may feel confused if the conclusions from different studies are conflicting. Authors could give a short explain or descript the condition of studies briefly, as you can explain your detailed analysis at the Discussion.

Response to comment 5: The reviewer has a point. However, given that section 3.2 describes “condition of included studies” in details (e.g., species, treatment doses, frequency, treatment duration, and outcome measures), reporting on these conditions once again in subsequent sections would be a sort of redundancy. Therefore, readers are encouraged to check Supplementary Table 1 for further details.

  1. In line 251, I think that the ”nitrogen species” is not current. It should be “reactive nitrogen species (ROS)”. Besides, the “NS” in line 254 and line 261 should also be modified to “RNS”. And in Figure 4, it is not proper to ignore RNS (or NS?) cause it has been describe before.

Response to comment 6: Thank you for this comment. To avoid confusion, we stick to “reactive nitrogen species (ROS)” since they represent the most common and most harmful form of free radicals. As the reviewer recommended, “NS” was replaced with “RNS”, (page 9, lines 318-328).and we included “RNS” in Figure 4 as well (page 10).

  1. In line 252, what does the “they” means? Does it mean “the free radicals such as ROS and RNS”? I think it should be clear.

Response to comment 7: Yes, we modified the text by replacing “they” with “ROS/RNS” for more clarity (page 9, lines 319).

  1. In 3.6, the last paragraph (line 259-269), the mechanism of antioxidant effects of 5-ALA should be stressed. For example, you can explain the relation between 5-ALA and the antioxidant factors such as HO-1 in that paragraph briefly. That will make reader more easily to understand.

 Response to comment 8: As suggested by the reviewer, we briefly highlighted on the possible antioxidant properties of 5-ALA (page 9, lines 298-305).

Discussion:

  1. In line 306-307, it is better to descript the content of the “improvement” detailed.

Response to comment 9: The term “improvement” was replaced with the exact nature of change “an increase in ADG” (page 11, lines 377-378).

  1. In the line 362, the sentence “cortisol hormone were measured only in single studies”, I think it should be confirmed whether there need to cite references.

  Response to comment 10: The relevant references were added (page 12, lines 442).

Minor concerns:

1.Line 148: replace “chitooligosccharide” to “chito-oligosccharide”.

This word has been modified accordingly (page 5, lines 157).

2.Line 149: It should have a blank at the beginning of the paragraph.

The statement has been modified accordingly (page 5, lines 159).

3.Line 239: replace “5ALA” to “5-ALA”

This word has been modified accordingly (page 9, lines 306).

4.Line 330: replace “two hr” to “2 hr”. Line 332: replace “three hr” to “3 hr”. I suggest that it is proper to use Arabic Number uniformly in this manuscript.

As recommended, Arabic numbers were uniformly used in the entire manuscript.

5.Line 331: replace “IL 6” to “IL-6”.

This word has been modified accordingly (page 11, lines 402).

Reviewer 2 Report

The use of 5-ALA as a dietary supplement is a non-traditional method which was tested in some animal species. Nevertheless, most of published papers evaluated in this review are made by one group of authors and as was stated (“within the context of the detected methodological flaws and the obvious variations in procedures, the spotted significant effects cannot be fully attributed to use of 5-ALA……”) the quality of evaluated study was not very good.

The materials and methods describe in the details the systematic review and numbers of evaluated papers. The description of quality criteria is quite long. Three of criteria was not fulfil in any of the evaluated study and one in all studies, but the description shows all possibilities even if they were not used. Almost the same results of quality criteria evaluation are described three times in the results (text, figure 2, figure 3, ). This part should be described more consistently and shorter.

The results describe the effect of 5-ALA supplementation in different species, but the authors did not pay attention to possible differences in metabolism (ruminants x monogastric animals x birds). I recommend to evaluate the results of the experiments also according the possible differences in species. Table 1 is quite large and contain a lot of information; nevertheless, not in clear schematic form. I recommend to rewrite the table according the animal species and instead of description of all parameters which were measured use individual column (i.e. – growth performance, production parameters, blood parameters, immune function…..) in which will be shown by +/ 0 / - / nd if these parameters were positively / no effect/ negatively influenced or was not measured (nd – not determined).

Line 211-212 – “…..,5-ALA improved Fe concentration in 11 of 15 results.” Please characterise better iron concentration (blood, plasma….)

The mode of action describing in the chapter is questionable. It is well known, that 5-ALA is synthesised in the body from glycine and succinyl-CoA with the help of enzyme ALA-synthase. This enzyme is rate-controlling; heme acts as a negative regulator of the synthesis of ALAS-1 and also the amounts of iron in the body play an important role. From this point of view, it is not clear why the synthesis of 5-ALA is inadequate and the supplementation is necessary (the substrate for the synthesis are freely available in the body or could be easily synthesised). Are the authors able to specify on the basis of evaluated papers in which situation the supplementation is recommendable / effective?

The mechanism of action which is described in the paper shows, that 5-ALA stimulates heme oxygenase-1, which is an enzyme starting heme degradation. It is at least strange if we supposed that synthesis of 5-ALA is regulating step for heme synthesis. This indicated also the authors (line 293 “..it has a role in heme synthesis through which, heme or hemoglobin content improves”).  The detected effect on oxidative stress is not uniform, but the described mechanism (figure 4) shows mainly anti-oxidant and antiinflammatory effect.  Are the authors able to explain these opposite functions of 5-ALA for heme synthesis / degradation?

The authors described also some combinations of 5-ALA with other additives. Mainly with vitamin C and some antibiotics. Nevertheless, they pay to this area very little attention. Concerning the combination with vitamin C they conclude “We can conclude that some species might benefit better than others from 5-ALA combinations”. Is it really conclusion? It says nothing. What is the expected effect of vitamin C supplementation? What about the reason and the effect of other combinations?

Author Response

Reviewer 2

Comments and Suggestions for Authors

  1. The materials and methods describe in the details the systematic review and numbers of evaluated papers. The description of quality criteria is quite long. Three of criteria was not fulfil in any of the evaluated study and one in all studies, but the description shows all possibilities even if they were not used. Almost the same results of quality criteria evaluation are described three times in the results (text, figure 2, figure 3, ). This part should be described more consistently and shorter.

Response to comment 1: Thank you for the reviewer’s concern about clarity of the text. We tried our best to make this section briefer both in the Methods and Results. In order to avoid reporting one every single study and inflate the text, we have to keep Figure 2 (which details every risk of bias for every single study) and Figure 3 (which portrays overall risk of biases in the whole review). This is also consistent with the PRISMA guidelines for reporting on systematic reviews.

  1. The results describe the effect of 5-ALA supplementation in different species, but the authors did not pay attention to possible differences in metabolism (ruminants x monogastric animals x birds). I recommend to evaluate the results of the experiments also according the possible differences in species. Table 1 is quite large and contain a lot of information; nevertheless, not in clear schematic form. I recommend to rewrite the table according the animal species and instead of description of all parameters which were measured use individual column (i.e. – growth performance, production parameters, blood parameters, immune function…..) in which will be shown by +/ 0 / - / nd if these parameters were positively / no effect/ negatively influenced or was not measured (nd – not determined).

Response to comment 2: We are grateful for your help simplifying this table. According to this comment, we have restructured Table 1 to report on major outcome indices according to type of species in brief—effects are shown by +/ 0 / - / nd if these parameters were positively / no effect/ negatively influenced or was not measured (nd – not determined). We included the stuffy table as a supplement in case readers would require more information (page 7).

  1. Line 211-212 – “…..,5-ALA improved Fe concentration in 11 of 15 results.” Please characterise better iron concentration (blood, plasma….)

Response to comment 3: In two studies iron was measure in plasma and in the rest of the studies iron was measured in serum. Accordingly, we noted in the text that iron status refers to iron levels in the serum unless indicated otherwise (page 7, lines 227-228).

  1. The mode of action describing in the chapter is questionable. It is well known, that 5-ALA is synthesised in the body from glycine and succinyl-CoA with the help of enzyme ALA-synthase. This enzyme is rate-controlling; heme acts as a negative regulator of the synthesis of ALAS-1 and also the amounts of iron in the body play an important role. From this point of view, it is not clear why the synthesis of 5-ALA is inadequate and the supplementation is necessary (the substrate for the synthesis are freely available in the body or could be easily synthesised). Are the authors able to specify on the basis of evaluated papers in which situation the supplementation is recommendable / effective?

Response to comment 4: Thank you so much for such deeply integrated and enlightening comments. We reported on the mechanism of 5-ALA synthesis in the text in this revised version (page 2, lines 66-67). According to the available studies, use of 5-ALA as a supplement is supposed to bring positive effects (e.g., improving blood iron status) rather than to make up for inadequate internal production (the latter was never mentioned in any study, which is consistent with what the reviewer mentioned in comment 4). In a specific reply to the question about situations in which the supplementation is recommendable / effective, this version has been supplemented with a paragraph detailing these conditions (page 11, lines 405-414).

  1. The mechanism of action which is described in the paper shows, that 5-ALA stimulates heme oxygenase-1, which is an enzyme starting heme degradation. It is at least strange if we supposed that synthesis of 5-ALA is regulating step for heme synthesis. This indicated also the authors (line 293 “..it has a role in heme synthesis through which, heme or hemoglobin content improves”).  The detected effect on oxidative stress is not uniform, but the described mechanism (figure 4) shows mainly anti-oxidant and antiinflammatory effect.  Are the authors able to explain these opposite functions of 5-ALA for heme synthesis / degradation?

Response to comment 5: We agree with the reviewer, the mechanism did not address various aspects of 5-ALA action. In this revised version, 5-ALA role in heme synthesis / degradation were further illustrated (page 8, lines 277-305; 336-340).

  1. The authors described also some combinations of 5-ALA with other additives. Mainly with vitamin C and some antibiotics. Nevertheless, they pay to this area very little attention. Concerning the combination with vitamin C they conclude “We can conclude that some species might benefit better than others from 5-ALA combinations”. Is it really conclusion? It says nothing. What is the expected effect of vitamin C supplementation? What about the reason and the effect of other combinations?

Response to comment 6: Thank you for this important comment. Not reporting on effects of other alternative supplements may cause confusion for the readers. In this regard, we have included a supplementary table reporting on effects of 5-ALA alone, comparison treatments alone, and combination of 5-ALA with other treatments. We also briefly compared between effects of 5-ALA and other treatments in the text (page 8, lines 252-275). We hope that reviewer 2 finds this section properly addressed in this revision.

Reviewer 3 Report

The 5-aminolevulinic acid (5-ALA) raises many emotions as to its effectiveness as an animal feed additive. Standard procedures and outcome measures are needed to confirm the benefits of 5-ALA. I consider the selection criteria for the cited works to be reasonable and correct. The figures are well developed and provide valuable information.

In the conclusion of the publication, could the authors indicate recommendations for animal nutrition practice? What amounts of 5-ALA and in what composition can they be recommended for cattle, pigs and poultry?

Specific comments

Line 4 -  A systematic review – better – A review,

Line 28, 57 and others -  5-Aminolevulinic acid – better  5-aminolevulinic acid,

Line 51 – FePO4 – should be – FePO4,

Line 152 – add – (ADFI),

Author Response

Reviewer 3

Comments and Suggestions for Authors

The 5-aminolevulinic acid (5-ALA) raises many emotions as to its effectiveness as an animal feed additive. Standard procedures and outcome measures are needed to confirm the benefits of 5-ALA. I consider the selection criteria for the cited works to be reasonable and correct. The figures are well developed and provide valuable information.

In the conclusion of the publication, could the authors indicate recommendations for animal nutrition practice? What amounts of 5-ALA and in what composition can they be recommended for cattle, pigs and poultry?

Thank you so much for such a vital comment. The conclusion of this modified version provides recommendations for use of 5-ALA in animal industry e.g., to corrects anemia in little pigs, increase egg production and quality, and improve milk iron and protein content. Based on reported doses, we provide a range of effect doses in different species (page 12, lines 449-454).

Specific comments

Line 4 -  A systematic review – better – A review,

The text was modified as reviewer recommended (page 1, line 4).

Line 28, 57 and others -  5-Aminolevulinic acid – better  5-aminolevulinic acid,

The statement was modified accordingly (page 1, line 30).

Line 51 – FePO4 – should be – FePO4,

The word was modified accordingly (page 2, line 58).

Line 152 – add – (ADFI)

Based on comments of Reviewer 2, we modified the tables and text in this version. Therefore, the average daily feed intake (ADFI) was used only once indicating no need for its abbreviation.

Round 2

Reviewer 2 Report

The corrections performed are sufficient. I have no more recommendations.

Minor notes:

Line 22 (abstract) -

“…. status and immunity were most responsive to 5-aminolevulinic.” Add “acid”

Figure 1:

Full text assessed for eligibility (n =..)   value is missing

Author Response

Reviewer 2

Minor notes:
Q1: Line 22 (abstract) -
“…. status and immunity were most responsive to 5-aminolevulinic.” Add “acid”

In this revision, the missing word “acid” was added (page 1; line 24)

Q2: Figure 1: Full text assessed for eligibility (n =..)   value is missing

 In this revision, the missing value in Figure 1 was added (page 4).
